# Maternal Hyperhomocysteinemia Disturbs the Mechanisms of Embryonic Brain Development and Its Maturation in Early Postnatal Ontogenesis

**DOI:** 10.3390/cells12010189

**Published:** 2023-01-03

**Authors:** Dmitrii S. Vasilev, Anastasiia D. Shcherbitskaia, Natalia L. Tumanova, Anastasiia V. Mikhel, Yulia P. Milyutina, Anna A. Kovalenko, Nadezhda M. Dubrovskaya, Daria B. Inozemtseva, Irina V. Zalozniaia, Alexander V. Arutjunyan

**Affiliations:** 1I. M. Sechenov Institute of Evolutionary Physiology and Biochemistry of the Russian Academy of Sciences, 194223 St. Petersburg, Russia; 2D.O. Ott Research Institute of Obstetrics, Gynecology and Reproductive Medicine, 199034 St. Petersburg, Russia

**Keywords:** homocysteine, electron microscopy, neurotrophins, semaphorin, caspase, matrix metalloproteinase, neocortex, hippocampus, neurodegeneration, rat

## Abstract

Maternal hyperhomocysteinemia causes the disruption of placental blood flow and can lead to serious disturbances in the formation of the offspring’s brain. In the present study, the effects of prenatal hyperhomocysteinemia (PHHC) on the neuronal migration, neural tissue maturation, and the expression of signaling molecules in the rat fetal brain were described. Maternal hyperhomocysteinemia was induced in female rats by per os administration of 0.15% aqueous methionine solution in the period of days 4–21 of pregnancy. Behavioral tests revealed a delay in PHHC male pups maturing. Ultrastructure of both cortical and hippocampus tissue demonstrated the features of the developmental delay. PHHC was shown to disturb both generation and radial migration of neuroblasts into the cortical plate. Elevated *Bdnf* expression, together with changes in proBDNF/mBDNF balance, might affect neuronal cell viability, positioning, and maturation in PHHC pups. Reduced *Kdr* gene expression and the content of SEMA3E might lead to impaired brain development. In the brain tissue of E20 PHHC fetuses, the content of the procaspase-8 was decreased, and the activity level of the caspase-3 was increased; this may indicate the development of apoptosis. PHHC disturbs the mechanisms of early brain development leading to a delay in brain tissue maturation and formation of the motor reaction of pups.

## 1. Introduction

Prenatal hyperhomocysteinemia (PHHC) leads to the toxic effect of the amino acid homocysteine (Hcy) on both mother and embryo. It causes the disruption of placental blood flow and can produce serious disturbances in the formation of the brain in offspring. In the recent literature, there are some scarce descriptions of the structural abnormalities induced by PHHC, mainly the various observations of neuronal death in forebrain structures [1,2]. Recently, we have shown the neuroinflammation processes to be also involved [3,4]. They were presented as the gliosis development under the essential increase of the proinflammatory cytokines content. Our results are consistent with the conclusions of other authors [5,6,7]. However, there are many other molecular and structural abnormalities described in different models of the disturbances in the embryonal period of the brain development. They might be also involved in the molecular and cellular mechanisms of brain development pathogenesis during PHHC and should be studied in this model.

The delay in brain tissue formation and maturation is a common nonspecific complication of brain development, observed in a different model of brain damage including hypoxic and toxic stress (see [8] for review) and hyperhomocysteinemia (HHC) [9]. Previously, we observed the same features of the developmental delay in cortical and hippocampus tissue of rat pups subjected to PHHC [3,4]. However, the mechanisms leading to the delay in brain tissue development and maturation are poorly understood and have never been studied in the PHHC model. 

Migration of the neuroblasts is one of the fundamental mechanisms determining the final shape and function of the nervous system. Environmental toxins or distinct genetic mutations can disturb neuronal migration resulting in abnormal cortical stratification and neuronal positioning. Defects in neuronal migration often produce the misplacement of a neuronal population leading to the death of specific neuronal cells, increasing the risks of the development of neurological disorders, including cortical dysplasia, mental disability, and epilepsy [10]. In recent studies, there has been considerable emphasis on the mechanisms of neuronal migration and cell guidance. However, little is known about the molecular mechanisms that disturb the migration and regulation of the migration onset timing. The balance between preventing and promoting migration is crucial for the position of cells; the maturation of correct neuronal types at a precise time makes it possible the establishment of functional connections in the brain.

Neurotrophic factors NGF and BDNF, as well as their receptors, take an active part in the functioning of the “mother-placenta-fetus” system during pregnancy. Neurotrophins are not only transported from the mother’s body to the fetus, overcoming the utero–placental barrier, but they are also produced by the placenta itself for the growth and development of the fetus [11,12]. Thus, as a member of the neurotrophin family, BDNF plays an important role in maintaining brain homeostasis, neurogenesis, and the modification of morphological and functional aspects of synaptic plasticity. NGF, in turn, is necessary for the growth of neurites, the development and survival of cholinergic neurons in the central nervous system, as well as for the regulation of neuronal functions [13]. BDNF, in association with TrkB, plays a positive role in regulating the migration of ventricular progenitor cells [14] and cerebellar granule cells [15]. Pro-forms of neurotrophins form a signaling complex with sortilin and the p75 neurotrophin receptor (p75NTR) and thereby mediate the initiation of apoptosis [16,17]. Recent studies show that the BDNF precursor, proBDNF, and the NT3 precursor, proNT3, act similarly to proNGF by binding to sortilin to induce neuronal apoptosis [18,19,20]. Thus, proneurotrophins often play opposite roles to mature neurotrophins in controlling neuronal survival and synaptic activity. Therefore, the balanced production of neurotrophins is crucial during the formation of the fetal brain.

In normal pregnancy, the total BDNF level increases in fetal umbilical cord blood during the progress of gestation [21]. At the same time, in the blood serum of pregnant women, the level of BDNF was also significantly induced in the second and third trimester compared to the first trimester of pregnancy [12,22]. Such an increase can be explained by the increased transplacental transport of BDNF, necessary for the fetal nervous system formation, as well as a gradual transition to the independent production of neurotrophic factors by the fetus and placenta. In a model of cultured mouse embryos, TrkB and BDNF are detected at early embryonic stages in trophectoderm cells [23]. In the rat CNS, BDNF immunoreactivity was detected since E13, especially in the neocortical subplate and developing neuroblasts of the cortical plate. TrkB is also expressed in the rat brain during early embryonic development at E13-14 [24].

Normally, the expression of p75NTR and TrkA is found in mouse embryos in the internal mass of the blastocyst and is absent in the trophoblast. At the same time, p75NTR expression precedes the beginning of TrkA expression [25]. It was also noted that in mice, TrkA mRNA could be detected for the first time in the brain at E8.5 [24].

For some of the earliest migration events of the cells that form the nervous system, semaphorin (SEMA) signaling is important. For example, during the development of the CNS, SEMAs regulate the migration of a number of neuron types, including cerebellar granular neurons [26], cortical neurons [27], and GABAergic interneurons [28]. In recent years, SEMAs and their receptors (neuropilins and plexins) have attracted the attention of researchers because of their high pleiotropic functions. Their impact on the formation of the hippocampus, which plays a crucial role in memory and learning, is especially pronounced, since it is the main target for neurodegeneration [29]. Lack of SEMA3E/PlexinD1 signaling leads to further changes in postnatal and adult hippocampal formation such as numerous misdirected ectopic mossy fibers. SEMA3E-deficient mice were found to have an abnormal presence of ectopic mossy fibers and synaptic endings in the granule cell layer, as well as impaired proliferation and increased neural excitability in the dentate gyrus [30].

Apoptosis plays an important role during the normal development of the mammalian nervous system and is observed in populations of developing nerve progenitor cells, differentiated post mitotic neurons, and glial cells. Depending on the time, two waves of cell death can be distinguished during pre- and postnatal development. An early wave of apoptosis was observed in proliferative zones, where morphogenetic signals such as BMP, Wnt, FGF, and Sonic hedgehog proteins play an important role [8,31]. The key mediator of apoptosis is Caspase-3, which is universally expressed in the developing nervous system. It has been shown that PHHC induces DNA fragmentation, increases the level of p53 mRNA, and also leads to a decrease in the content of the antiapoptotic protein Bcl-2 in the brain tissue of newborn rats [32].

In this present study, we have investigated all these parameters in descendent male rats from mothers subjected to HHC during 4–21 days of the gestation.

## 2. Materials and Methods

### 2.1. Experimental Animals

Nulliparous female Wistar rats (3–4 months old) were obtained from the Rappolovo Animal Center, St. Petersburg, Russia. The animals were maintained on a 12/12 h light/dark cycle at a constant room temperature with free access to a 20% (*w/w*) protein commercial chow and clean drinking water throughout the study. The experiment involved female rats with an established normal 4-day estrous cycle, without signs of illness and abnormal behavior. All experimental protocols were performed in accordance with guidelines of the Declaration of Helsinki and approved by the Institutional Ethics Committees of D.O. Ott Research Institute of Obstetrics, Gynecology, and Reproductive Medicine (protocol code 88 as of 8 December 2017) and the I.M. Sechenov Institute of Evolutionary Physiology and Biochemistry RAS (protocol code 3/2020 as of 18 March 2020).

### 2.2. Chronic Methionine Treatment

The first day of pregnancy was considered the day after the detection of sperm in the vaginal smear of the female rat. Then, the animals were daily administrated with a 0.15% aqueous solution of L-methionine (Merck, Darmstadt, Germany, 0.10–0.15 g per animal) per os from the fourth day of pregnancy (E4) until delivery [33]. This modeling method allowed achieving serum Hcy concentrations in rats similar to those described for the serum of patients with mild HHC. The chronic administration of methionine at this dose to pregnant rats caused an increase in the level of Hcy not only in the mother’s blood but also in the blood and brain of the fetus, as described in detail previously [33]. In the same period, control animals additionally received 1 mL of water using a probe. On E14 and E20, these groups of rats were decapitated, and fetuses were removed from their mothers in both groups. Embryonic brains were used for analysis. Another group of pregnant rats gave birth naturally. The day of birth was determined as P1. Pups were decapitated at postnatal days 5 (P5), 14 (P14), and 20 (P20). The treatment paradigm followed for the study is illustrated in Figure 1a.

### 2.3. Offspring Somatic Development

Somatic development was assessed in control (n = 36) and PHHC (n = 49) pups on P1-P21 by measuring body weight and recording eye opening time.

### 2.4. Animal Behavior

To analyze the effects of PHHC on the functional state of brain structures and development of animal motor activity, we have used the following specific tests: body righting on P1-P8; whisker placing on P1-P15; rotating grid test on P1-P15; locomotor activity on P17-P19 and P23-P24 (Figure 1b). Neurodevelopmental testing was conducted at the same time daily. The rat pups were directly under a heated lamp during testing because the younger pups were less capable of thermoregulation. Male Wistar rats from ten litters, comprising control (n = 36 from 5 litters) and PHHC (n = 49 from 5 litters) pups, were used.

#### 2.4.1. Body Righting

In the body righting test, the rat pup was carefully placed on its back and its motor response was evaluated according to the following criteria: 1—no response (for lying on its back); 2—weak reaction (for lying on the left or right side); 3—incomplete reaction (for the pup’s ability to turn the body, but in the wrong position; for example, a leg left under the body of the animal); 4—complete response (for successful righting and appropriate posture). The maximal time taken for the animal to demonstrate a full extension response was 60 s.

#### 2.4.2. Whisker Placing

For the whisker placing test, the animal was lifted by the tail and the whisker area was touched with the sharp end of a horizontally oriented pencil. The reaction of forelimb placing was assessed during 60 s by a four-point scale: 1—no reaction (for lack of success), 2—chaotic, weak raising of the limb without contact with the stick, 3—head rotational movements and raising the limb to the support with one forelimb placing, 4—precise stretching the snout towards the stick and accurate elevation of both fore-limbs on the support.

#### 2.4.3. Rotating Grid

The rat pup was placed on a horizontal grid (100 × 100 mm, cell size 5 × 5 mm, metal thread diameter 1 mm) so that the pup’s head was against the direction of the grid movement, which was carried out at a speed of 2 rpm. The angle of grid rotation was recorded when the rat fell on a soft mat.

#### 2.4.4. Locomotor Activity

The motor activity of rat pups was analyzed for 60 s in an open field (500 × 500 mm, divided into 25 equal squares) by recording the number of squares crossed during running. The floor of the chamber was cleaned after each animal with a 50% ethanol solution

### 2.5. Analysis of Migration and Positioning of Cortical Neurons Generated on E14 or E18

The same protocol of the cell position analysis was performed as we had used previously to analyze neuroblast migration on the model of prenatal hypoxia; described in [34]. On the 14th or 18th day of gestation, pregnant female Wistar rats from control (n = 4) and HHC (n = 5) groups were subjected to single injection with 5′-ethynyl-2′-deoxyuridine (EdU)—a DNA-replication marker. One single intraperitoneal injection in a dose of 25 mg of EdU in 0.5 mL of saline was made to pregnant females to label cells generated in the proliferative zone of fetuses’ cerebral bladder walls. Cells labeled on E14 are destined to the lower cortical layers (V and VI), and cells labeled on E18—to the superficial cortical layers (II-III). We analyzed the positioning of neuronal cells generated on E14 or E18 by methods described in [34]. The pups were sacrificed and decapitated on the fifth day after birth when the number of labeled cells is known to be optimal for the quantitative analysis [34]. The brain tissue was fixed by immersion into 10% formalin in phosphate-buffered saline (PBS, pH 7.4) during 5 days and cryoprotected in 20% sucrose in PBS. The blocs of the forebrain were frozen and sectioned on cryomicrotome Leica CM 1510S (Leica Microsystems, Wetzlar, Germany) to prepare a series of slices at the coronal plane. A sequence of 15 µm sections with 30 μm between them (10 sections per animal) were randomly selected and used for the visualization of the labeled cells. The EdU tags in the nuclei of the cells were labeled using the Click-iT EdU Alexa Fluor 488 Imaging Kit (Thermo Scientific, Waltham, MA, USA) according to the manufacturer’s protocol. The nuclei of the cells were counterstained with Hoechst 33342 to reveal cortical stratification. This made it possible to count the total number of cells. In selected slices, cells were labeled by specific antibodies to the neuronal marker Fox3 (ab104224; Abcam, Bristol, UK; dilution 1:1000) to prove that the EdU-positive cells were neurons. Slices were incubated in PBS with blocking solution containing 2% bovine serum albumin, 0.3% Triton X-100 (Merck, Darmstadt, Germany), then were incubated overnight at 4 °C in the same blocking serum with rabbit polyclonal antibody to Fox3/NeuN (ab104224; Abcam, Bristol, UK; dilution 1:1000). After rinsing, the slices were incubated in phycoerythrin (PE)-tagged secondary anti-rabbit antibodies (ab7007, Abcam, 1:200) for 1 h at 37 °C. For the negative control, some slices proceeded with the same protocol but without the primary antibodies, and nonspecific immunoreactivity was not observed. Microscopy was performed on a Leica DMR microscope with a Leica SP5 confocal scanner (Leica Microsystems, Germany). Alexa Fluor 488 and PE were excited at 488 nm wavelength and Hoechst 33342 at 350 nm. Emissions of Alexa Fluor 488, PE, and Hoechst 33342 were observed in the sequent diapasons: 496–537 nm, 652–690 nm, and 430–461 nm wavelengths, respectively. We analyzed the object as immune-positive if it was more than 300% of the background signal. The brightness of the nuclei and bodies of the cells were measured using Video TesT-Morphology software program (Video TesT, Saint Petersburg, Russia).

Brain slices of 0.20 mm from Bregma [35] (see Figure 2a) were used to analyze the pattern of migration and positioning of E14- or E18-generated cortical neurons. The number of EdU-labeled cells was calculated within a 500 μm-wide area of the parietal cortex from the layer I to VI. A ratio of the number of the EdU-positive cells to the total number of Hoechst-labeled cells was calculated in each analyzed cortical slice. The comparison of the groups was performed by this ratio. The ratio of the numbers of EdU-positive cells placed within the superficial (the layers II-III) and lower (V-IV) layers were also calculated and compared in control and PHHC pups.

### 2.6. Electron Microscopy

CA1 and cortical tissue were our main area of interest as they are known to be involved in the mechanism of memory abilities. The areas of interest are presented in Figure 2a,b. On P5, P14, and P20, the ultrastructure of the parietal cortex and dorsal hippocampus was studied in the blocks of brain tissue, starting respectively at the level 0.20 mm and 4.5 mm from Bregma [35]. Three pups per group was analyzed in PHHC and control groups. After transcardial perfusion by a fixative containing 1% of glutaraldehyde and 1% formaldehyde in 0.1 M PBS, pH 7.4, brain tissue was additionally fixed in 1% OsO_4_. Brain tissue was stained with uranyl acetate, dehydrated, and embedded in Araldite by the protocol described in our previous paper [36]. Ultra-thin sections of 500 Å thickness were made using Leica ultramicrotome (Leica Microsystems, Germany) and analyzed on an FEI Tecnai Spirit V2 (FEI, USA) transmission electron microscope. The structural features of both cortical and hippocampus cells and neuropile were analyzed.

### 2.7. Quantitative PCR (RT-qPCR)

For this assay, whole embryonic brains were used on E14 and E20. Total RNA was extracted from brain samples using the ExtractRNA reagent (Evrogen, Moscow, Russia). For the reverse transcription reaction, 2 µg of total RNA, oligo-dT-primers, and 9-mer random primers and 200 units of MMLV reverse transcriptase (Evrogen, Moscow, Russia) were taken. Subsequently, the samples were diluted eightfold for PCR reactions. qPCR was carried out using 0.5 units of TaqM-polymerase (Alkor Bio, St. Petersburg, Russia), specific forward primers (200 nM), reverse primers (200 nM), and probes (100 nM) (see Table 1) in a volume of 6 µL. A 0.8 µL of cDNA sample was taken per PCR reaction. Multiplex systems were used for gene analysis: *Trkb* + *P75ntr*, *Vegfa* + *Kdr*, *Ywhaz* + *Pgk1*. In the case of *Bdnf*, *Ngf*, *Vegfb,* and *Trka* expression analysis, 50X SYBR Green I intercalating dye (Evrogen, Moscow, Russia) was used. PCR reactions were conducted in a C1000 Touch thermal cycler combined with a CFX384 Touch™ Real-Time PCR Detection System (Bio-Rad, Hercules, CA, USA) in triplet, simultaneously with no template and no reverse transcription control samples. Melting curve analysis was also performed for *Bdnf*, *Ngf*, *Vegfb,* and *Trka*. The relative expression of *Bdnf*, *Ngf*, *Vegfb*, *Trka*, *Trkb, Vegfa, Kdr,* and *P75ntr* genes was normalized against the relative expression of *Ywhaz* and *Pgk1* genes. The results were processed with the 2^−ΔΔCt^ method [37] normalized against the relative expression of *Ywhaz* and *Pgk1* genes.

### 2.8. Western Blot Analysis

For this assay, whole embryonic brains were used on E14 and E20. Brain tissue was homogenized on ice 1:2 (*w/v*) in a 0.01 M PBS buffer (pH 7.4). Tissue homogenates were then centrifuged (16,000× *g*, 20 min, 4 °C), and only supernatants were used. Determination of the protein concentration in the homogenates was carried out using bovine serum albumin as a standard according to the Bradford method. For the immunoblot run, equal amounts of protein (50–80 μg as recommended for each antibody) from each sample were separated by SDS-PAGE, and Western blot was performed as previously described [4]. Immunoblotting was carried out with primary antibodies against NGF (ab52918, Abcam, USA; dilution 1:1000), BDNF (ab108319, Abcam, USA; dilution 1:1000), SEMA3E (MAB3239, R&D Systems, Minneapolis, MN, USA; dilution 1:1000), Caspase-3 (9662S, Cell Signaling Technology, Danvers, MA, USA; dilution 1:1000), Caspase-8 (9746S, Cell Signaling Technology, Danvers, MA, USA; dilution 1:1000), and VEGFA (ab46154, Abcam, USA; dilution 1:1000). On the next day, blots were incubated with a secondary peroxidase conjugated antibody (#1706515 or #1706516, BioRad, Hercules, CA, USA; dilution of 1:3000). For SEMA3E, we used goat anti-mouse Ig peroxidase conjugated antibody (HAF007, R&D Systems, Minneapolis, MN, USA; dilution 1:1000). Densitometric analysis of each protein was performed using the Image Lab™ 5.2.1 software (Bio-Rad, Hercules, CA, USA). Based on the existing recommendations for the target protein content normalization [42], the obtained band optical density was normalized to the content of total protein in the gel determined using the stain-free technology according to the manufacturer’s instructions (BioRad, Hercules, CA, USA).

### 2.9. Gelatin Zymography

For the analysis of matrix metalloproteinase-2 (MMP-2) activity, fetal brain homogenates were prepared on ice 1:2 (*w/v*) in a 0.01 M PBS buffer (pH 7.4). Samples were then centrifuged (16,000× *g*, 20 min, 4 °C). Supernatants obtained were used for further analysis. The fetal brain samples for zymography were standardized according to protein content and volume (15 μg/27 μL). Gelatin (1 mg/mL, Sigma Aldrich, St Louis, MO, USA) was copolymerized with 10% SDS-PAGE (Bio-Rad, Hercules, CA, USA). Gels were run under nonreducing and nondenaturing conditions. Gels were briefly washed with dH_2_O three times for 5 min. The SDS was removed by washing the gels in 37 °C Tris buffer (50 mM Tris, 5 mM CaCl_2_, 2.5% Triton X-100, pH 7.6) three times for 30 min. After the activation, gels were incubated (18 h, 37 °C) in Tris-HCl buffer (50 mM Tris-HCl, 0.2 M NaCl, 5 mM CaCl_2_, 0.02% Brij-35, 1 μM ZnCl_2_, pH 7.6). The MMP-2 activity in gels was visualized using Coomassie Brilliant Blue R-250 Staining Solution (BioRad, Hercules, CA, USA) and destaining in 10% acetic acid:25% ethanol solution. Gel visualization was performed using ChemiDocTM Touch Imaging system (Bio-Rad, USA). Analysis of band optical density was performed using the Image Lab™ 5.2.1 software (Bio-Rad, Hercules, CA, USA).

### 2.10. Caspase-3 Activity

For this assay, whole embryonic brains were used on E14 and E20. Samples containing 120 µg of protein were incubated at 37 °C for 10 min with the reaction buffer (20 mM HEPES, containing 0.1% CHAPS, 2 mM EDTA, and 5 mM DTT; pH 7.4). The reaction was initiated by adding 4 mM synthetic peptide Ac-DEVD-pNA (N-acetyl-Asp-Glu-Val-Asp p-nitroanilide) as substrate. The absorbance of the reaction mixture was recorded every 5 min for 35 min at 405 nm. The caspase-3 activity was determined as micromoles of pNA (the reaction product)/min/mg protein.

### 2.11. Oxyblot Analysis

Protein oxidation levels were detected using an OxyBlot™ Protein Oxidation Detection Kit (Merck Millipore, USA) according to the manufacturer’s protocol. Briefly, 5 μL (15 μg of protein) of sample was mixed with the reaction solution (5 μL of 12% SDS, and 10 μL of 2,4-dinitrophenylhydrazine derivatizing agent) and further incubated for 15 min at room temperature. The reaction was stopped by adding 7.5 μL of neutralization buffer followed by 1.5 μL β-mercaptoethanol. Carbonyl-derivatized proteins in the samples were loaded on 10% SDS-PAGE (TGS Stain-Free FastCast Acrylamide Kit, Bio-Rad, Hercules, CA, USA) and then transferred onto a PVDF membrane (Bio-Rad, Hercules, CA, USA). Membranes were blocked in 1% bovine serum albumin in TBS buffer (pH 7.5) containing 0.1% Tween-20 for 1 h at room temperature. Membranes were then incubated with a primary antibody (1:150) provided by the kit at 4°C overnight. This was followed by incubation with the corresponding HRP-conjugated goat monoclonal antibodies (Goat Anti-Rabbit Ig (H+L)-HRP; dilution 1:1000; BioRad, Hercules, CA, USA) for 1.5 h at room temperature. 2,4-dinitrophenyl-modified protein bands visualization was performed using ChemiDocTM Touch Imaging system (Bio-Rad, USA). Analysis of optical density was performed using the Image Lab™ 5.2.1 software (Bio-Rad, Hercules, CA, USA). 

### 2.12. Measurement of Lipid Peroxidation

Lipid peroxidation was measured in the E20 brain homogenates using the thiobarbituric acid (TBA) test. This method determines the concentration of malondialdehyde (MDA), which is the most important end-product of lipid peroxidation. For this assay, fetal brain homogenates were prepared on ice 1:2 (*w/v*) in a 0.01 M PBS buffer (pH 7.4). Tissue homogenates were then centrifuged (16,000× *g*, 20 min, 4 °C), and only supernatants were used in analysis. Two 300 µL glass tubes of 2% (*w/v*) orthophosphoric acid, 20 µL of sample, and 100 µL of 0.8% (*w/v*) TBA were added. The control sample contained distilled H_2_O instead of tissue homogenate. After boiling for 45 min, the probes were cooled to room temperature. To extract malondialdehyde, (MDA)-TBA adduct 300 µL n-butanol was added to the mixture. All samples were then centrifuged (3500× *g*, 15 min). The supernatant was moved to 1 cm cuvettes and optical density was measured using NanoDrop One spectrophotometer (Thermo Scientific, USA) at wavelengths of 532 and 580 nm. Results were expressed as μmol MDA/mg protein.

### 2.13. Statistical Analysis

All analyses were randomly performed in a blinded manner for all animal groups. In all experiments, each animal and sample were given a random number so the experimenters were not aware of the tested groups, which were only revealed during the statistical analysis. Statistical analysis was performed using the STATISTICA 10.0 software. The normality of the data was tested using the Shapiro–Wilk normality test. To verify the equality of variances, the Levene test was used. In behavioral studies, two-way analyses of variance were made for two independent groups on the day factor, with Student–Newman–Keuls post hoc analysis. In the case of unequal variances, the raw data were transformed into ln values. Locomotor activity data were assessed by one-way ANOVA with Student–Newman–Keuls Method post hoc analysis. In histochemical and neurochemical studies, variance was analyzed by the Mann–Whitney *U*-test and Student’s *t-*test. Values that were normally distributed and compared by parametric statistical methods are presented as mean ± standard error of mean (SEM). The data, the distribution of which did not obey the normal law and which were compared by nonparametric statistical methods, were presented as median (25th, 75th percentile). Values of *p* ≤ 0.05 were considered statistically significant.

## 3. Results

### 3.1. Body Weight (P1-P21)

There was a significant day, group, and their interaction effects for the body weight (Table 2). As shown in Figure 1a, PHHC pups tended to decrease in body weight compared to control ones, and on P6, P8-P10, P13-P15, P18, and P20-P21, this reduction was statistically significant (*p* ≤ 0.05). Eye opening in both control and PHHC rat pups occurred on P17.

### 3.2. Animal Behavior

#### 3.2.1. Body Righting (P1-P8)

There was a significant day, group, and their interaction effects for the body righting score (Table 2). Insufficient motor coordination did not allow newborn animals of both groups to fully implement turns during the performance of this gravitational reflex. However, control pups demonstrated more mature body righting response compared to PHHC pups, showing significant (*p* ≤ 0.05) differences (according to a three-point scoring system) from P1 to P5 (Figure 1b). Complete rotation was performed by 100% of control pups on P5 and only 55% of PHHC pups on P6.

#### 3.2.2. Whisker Placing (P1-P15) 

There was a significant day, group, and interaction effects for the body whisker placing score (Table 2). From P1 to P13, PHHC pups compared to control pups had a significantly lower (*p* ≤ 0.05) score of forelimb placing after whisker stimulation (Figure 1c). Complete fulfillment of the studied reaction in 100% of PHHC pups occurred on the 15th day i.e., three days later than in the control group.

#### 3.2.3. Rotating Grid (P1-P15) 

There was a significant day, group, and interaction effects for the rotating grid test (Table 2). Significant differences (*p* ≤ 0.05) in the maximal angle before falling of control and PHHC pups were seen from P8 to P15 (Figure 1d). On average, control pups were able to remain on a rotating grid up to 90 degrees on P7, and PHHC pups—on P10. The maximal angle before falling in the PHHC group reached was 139 to P15, while in the control group the maximal angle reached was 160 to P13.

#### 3.2.4. Locomotor Activity (P17-P19 and P23-P25) 

We assessed the locomotor activity of rat pups for three consecutive days immediately after opening their eyes, i.e., after P17, as well as one week after this event. There was a significant group difference for this test (F _1;11_ = 4.026, *p* ≤ 0.001). As seen in Figure 1e, immediately after opening the eyes, locomotor activity was significantly higher in control pups (*p* ≤ 0.05) and remained at the same level after a week (*p* ≤ 0.05). In PHHC pups, the locomotor activity was initially lower than that of the control, but after a week, it reached the level of the control pups (*p* ≤ 0.05).

### 3.3. Analysis of Migration and Positioning of Cortical Neurons Generated on E14 or E18

**Control.** Double labeling of cells with a neuron-specific marker Fox3 and EdU was performed to reveal the neuronal nature of cells generated in the periods of EdU labeling. The majority of the cells generated on E14 or E18 in both the control and PHHC group were shown to be Fox3-positive neurons (Figure 2c). The distribution of the EdU-labeled cells in cortical plate were shown to be depended on the time of cell labeling. In the case of the cell labeling on E14, about 79% of the EdU-positive cells were observed in the lower (V-VI) cortical layers (Figure 2d). The same time, about 80% of cells labeled on E18 were localized in the superficial (II-III) cortical layers (Figure 2f).

**PHHC.** In the parietal cortex of 5 day-old pups subjected to PHHC, we observed a decrease in the total number of EdU-positive neurons labeled on both E14 (*p* ≤ 0.01, Mann–Whitney U-test; see Figure 3a) and E18 (*p* ≤ 0.01, Mann–Whitney U-test; see Figure 3c) compared to controls. It can be suggested that PHHC causes a visible decrease in neuroblast division rate. In both PHHC animals and controls, the majority of cells labeled on E14 were destined to migrate in the V-VI cortical layers. In PHHC pups, the number of labeled neurons localized within the superficial layers of the parietal cortex was increased (*p* = 0.03, Mann–Whitney U-test) compared to controls (Figure 2e and Figure 3b). This can suggest that PHHC caused a failure of some neuroblasts to migrate into their proper position in the lower cortical layers. The majority of EdU-positive cells labeled on E18 were scattered within the superficial (II-III) cortical layers (Figure 2g) in both control and PHHC groups. However, in PHHC pups, the number of EdU-positive cells observed in the V-VI layers of the parietal cortex was increased (*p* = 0.01, Mann–Whitney U-test) compared to the control group (Figure 2g and Figure 3d). It can be suggested that these neuroblasts failed to migrate into their proper position in the II-III cortical layers.

It can be concluded the PHHC on E4-E21 disturbed both generation and radial migration of the neuroblasts in the parietal area of the cortical plate.

### 3.4. Electron Microscopy

#### 3.4.1. Neural Tissue Formation and Maturing in P5 Pups 

**Parietal cortex.** An electron microscopic study of the parietal cortex at P5 in PHHC pups revealed the features of a delay in neurogenesis and, most importantly, synaptogenesis compared control. The following indicators of tissue immaturity were observed in PHHC pups’ cortical tissue: increased volume of intercellular space and numerous growth cones. In contrast to control pups at this developmental stage, in PHHC pups, the nervous tissue of the cortex the volume of intercellular space was increased and might reach up to several microns (Figure 4a–d). The number of growth cones containing large light rounded growth vacuoles was increased in the neuropil of the cortex. There were vacuoles of various sizes and shapes (Figure 4d). In the neuropil of control pups, there were growth cones with small vesicles similar to synaptic ones, which accumulate on one of their poles (Figure 4a). In such places, flattening of the synaptic membrane was noticeable. Growth cones in PHHC pups were mainly found near neurons. In control pups, the narrow cytoplasm surrounding nucleus contains a complete set of cellular organelles: rough endoplasmic reticulum with ribosomes, the mitochondria, and the Golgi complex. In PHHC pups, the cytoplasm of neurons was not yet completely filled with organelles (Figure 4c), although a complete set of organelles is required for respiratory function and protein synthesis.

**CA1 area of hippocampus.** The neuropil of the hippocampus of control pups was filled with numerous growing processes, cut lengthwise and across. On P5, unmatured neurons with oval nuclei contained dispersed chromatin and were arranged in groups (Figure 4e) in both control and PHHC pups. In PHHC pups, they were more numerous than in control pups of the same age. In PHHC pups, poorly differentiated neurons were separated from each other by a large amount of intercellular space. An increased number of growth cones were found in the neuropil (Figure 4e,h). In the cytoplasm of some neurons, as shown in Figure 3g, a slight swelling of the endoplasmic reticulum and the Golgi complex was noted. Numerous ribosomes and lysosomes were scattered in the cytoplasm of neurons around the nucleus.

#### 3.4.2. Neural Tissue Formation and Maturing in P14 Pups

**Parietal cortex.** On P14, in the control rats, the process of further differentiation of neurons took place in the cortex. Ribosomes, the Golgi complex, and rough endoplasmic reticulum and mitochondria were observed in their cytoplasm. However, a large volume of intercellular space (Figure 5a) is still visible in the neuropil, and the growth cones (Figure 5b) are often found, especially near neurons. An increased number of glial cells with processes appeared, there was an intensified growth of dendrites and their processes, dendritic tubules were visible in the dendrites, and occasionally spines appeared but without a spine apparatus. During this period of dendritic growth in control rats, a rather low number of true synapses with accumulated synaptic vesicles in the presynaptic region and pronounced thickening of synaptic membranes were detected. PHHC rats showed a significant delay in neurogenesis and synaptogenesis compared to control. They demonstrated a less dense neuropil, increased volume of intercellular space, numerous growth cones, desmosomes contacts, and low number of the spines (Figure 5d) than in the control. Neurons at this stage of development were less differentiated than in the control. In their cytoplasm, as shown in Figure 5c, swollen cisterns of the endoplasmic reticulum and the Golgi complex were found at this stage of development.

**CA1 area of hippocampus.** In the hippocampus of control two-week-old rats, further maturation of neurons occurred. The neuropil became denser, although intercellular spaces and growth cones were still found (Figure 5e,f). Intercellular spaces were often located next to neurons, so axosomatic contacts are rare. In PHHC pups, there were few spines and less mature synapses than in the control. In the hippocampus of PHHC, a significant lag in neurogenesis and synaptogenesis was noticeable. In such rats, a less dense neuropil was observed with frequent intercellular space and growth cones. Neurons were less differentiated, with many free ribosomes in the narrow cytoplasmic rim around the cell nucleus. The Golgi complex and the endoplasmic reticulum were not enlarged (Figure 5g). There were much fewer spines, and the synapses (Figure 5h) were less mature than in the control.

#### 3.4.3. Neural Tissue Formation and Maturing in P20 Pups 

**Parietal cortex.** On P20, in the parietal cortex of control rats, there were a decrease in the volume of the intercellular space and a decrease in the number of growth cones (Figure 6a,b). In the neuropil, the number of glial cells with their processes increased. Neurons at this stage were differentiated cells containing a large oval nucleus with two nucleoli. Regarding the cytoplasm surrounding the nucleus of such cells, we observed a complete set of cellular organelles: rough endoplasmic reticulum and the Golgi complex, mitochondria, and ribosomes (Figure 6a). The number of dendrites with branches and dendritic tubules increased. On such branches of dendrites, immature dendritic spines appeared, but mainly without a spine apparatus. Myelinated fibers were not numerous at this stage. In the neuropil of the parietal cortex were observed both asymmetric axo-dendritic contacts and symmetrical axo-somatic synapses. Numerous contacts appeared on varicose thickenings of dendrites and axons. There were also many synaptic contacts in the glomeruli specific to the neuropil of the cortical regions (Figure 6b). As shown in this figure, a transversely cut dendritic process was surrounded by numerous axon terminals with mature synaptic contacts. During this period of development, a significant number of mature synapses were revealed with an accumulation of synaptic vesicles in the presynaptic region and with a pronounced thickening of the postsynaptic density. In cortex PHHC rats, a delay in cell differentiation, neurogenesis, and synaptogenesis was revealed. In the cytoplasm of many neurons, the expansion of the endoplasmic reticulum cisterns with abundant ribosomes and especially polysomes were noticeable (Figure 6c,d). In the neuropil of the cortex of such rats, there were preserved intercellular spaces and growth cones (Figure 6d). There were much fewer synaptic contacts than in the neuropil of the cortex of control rats, especially mature synapses with accumulation of synaptic vesicles in the presynaptic section and thickened postsynaptic density.

**CA1 area of hippocampus.** The maturation process of the hippocampus nervous tissue of PHHC rats was delayed compared to the control. In control rats, the intercellular space becomes much smaller; in neurons, a further process of differentiation occurs, the formation of the complete set of cell organelles. Axon terminals were placed around neurons, but axo-somatic contacts were still undeveloped. In the neuropil of the hippocampus, a large number of transversely and longitudinally cut axons were observed, although the contacts are still in the form of desmosomes (Figure 6e,f). Myelinated fibers at this stage of development were mainly absent in the hippocampus; a large number of immature spines were found. In the same period, in the hippocampus of PHHC rats, some retardation in the development of the nervous tissue was noticeable. In a small number of neurons, endoplasmic reticulum edema occurs in the cytoplasm, contacts in the form of desmosomes between processes were observed in the hippocampal neuropil, and there was still a small amount of intercellular space and growth cones (Figure 6g,h).

Thus, the study of the neurogenesis and synaptogenesis in the cortex and hippocampus during period P5-20 revealed a significant delay in the neural tissue development in PHHC rats.

### 3.5. Analysis of the Neurotrophin System Components in the Rat Fetal Brain

There were no significant changes in the expression of *Ngf*, *Trka*, *Trkb*, and *P75ntr* mRNAs in the brains of control group fetuses and fetuses that developed under conditions of elevated Hcy levels on E14 and E20 (Figure 7a). A statistically significant increase (Student’s *t-*test, *p* ≤ 0.05) in the level of *Bdnf* expression in the brain of fetal rats was observed only in the group with maternal HHC on E20 compared with the control. 

Analysis of BDNF levels in fetal brains revealed two bands with different molecular weights corresponding to the mature (~14 kDa, mBDNF) and pro-form (~29 kDa, proBDNF) neurotrophic factor (Figure 7b,d). During prenatal development, an increase in the proBDNF content by 30% from E14 to E20 was detected in the brain of rat fetuses from the control group (Figure 7b). In the PHHC group, a significant increment (Student’s *t-*test, *p* ≤ 0.05) in the level of proBDNF on E14 was revealed by 1.1 times compared with the control, as well as a decrease in mBDNF by 10%. At the same time, only a 1.2-fold increase in pro-form relative to control was noted in the fetal brain on E20.

When analyzing the NGF content in the fetal brain by the Western blot method, one band with a molecular weight of ~31 kDa was identified, which corresponded to proNGF (Figure 7c,d). The level of the pro-form of this neurotrophin in the control group of rat fetuses on E20 was statistically lower (Student’s *t*-test, *p* ≤ 0.001) than on E14. At the same time, there were no changes in the content of proNGF in the brains of control fetuses and fetuses that developed under conditions of elevated Hcy levels on E14 and E20.

### 3.6. Analysis of Caspase-3 and Caspase-8 in the Rat Fetal Brain

In both groups of rat fetuses, the active form (p17 fragment, 17 kDa) of Caspase-3 was detected in the brain only on E20, while its inactive precursor (procaspase-3, 35 kDa) was identified at both studied developmental stages (Figure 8a,d). There were no significant differences in the content of Caspase-3 fragments. However, in the fetuses of rats after PHHC, an increment (*p* ≤ 0.05) in the activity of Caspase-3 in the brain by 20% was detected on the E20 compared with the control (Figure 8b). 

When studying the content of Caspase-8 in the fetal brain, three fragments with different molecular weights were identified: procaspase-8 (57 kDa, p57), split Caspase-8 (43 kDa, ~p43/41) and active fragment (18 kDa, p18) (Figure 8c,d). At the same time, in the group with PHHC, we noted a statistically significant (Student’s *t-*test, *p* ≤ 0.05) decrease (by 1.25 times compared with the control) in the procaspase-8 content on E20 (Figure 8c). However, the content of other fragments on E14 and E20 did not differ from the control group.

### 3.7. Analysis of the Angiogenesis System Components in the Rat Fetal Brain

Analysis of the mRNA content of the angiogenesis system components showed that in the fetal brain on E14, the relative expression of the *Kdr* gene at the mRNA level was significantly reduced (by 1.17 times) compared with the control, but there were no significant changes in this marker on E20 (Figure 9a). There were also no significant changes in the expression levels of *Vegfa* and *Vegfb* mRNA in the brains of fetuses whose mothers were injected with methionine.

No statistically significant differences were found in the study of VEGFA content in the brain of rat fetuses. The relevant data are shown in Figure 9b,e.

A study of the SEMA3E content revealed its decrease in fetal brain tissue on E14 in PHHC (*p* ≤ 0.05) compared with the fetuses from the control group (Figure 9c,e). However, on E20, there were no significant changes in the content of this factor and the decrease was only a trend.

As a result of measuring the MMP-2 activity, its decrease was revealed during PHHC in the brain (Student’s *t-*test, *p* ≤ 0.01) of embryos on E14 (Figure 9d,e). In the fetal brains of rats from the PHHC group on E20, the decrease in the activity of this enzyme had only a trend nature.

### 3.8. Oxidative Modifications of Macromolecules

Our studies of oxidative stress markers showed no changes in oxidative modifications of proteins in the fetal brain during PHHC (Figure 10a,b). However, an increase in lipid peroxidation in the fetal brain on E20 was noted under the influence of PHHC, which was concluded on the basis of the malondialdehyde content data (Figure 10c).

## 4. Discussion

The observed decrease in the body weight of PHHC pups is quite consistent with the previously described facts of the limited intrauterine growth of offspring from mothers with high levels of homocysteine in the body [43,44]. At the same time, it is noted that higher concentrations of Hcy in the mother’s blood are associated with a small risk of weight loss in newborns [45], which was the case in our experiments. The observed small difference in newborn weight may be of little clinical significance to the individual; however, it may matter more at the population level. It is known that the growth of embryos depends on the normal functioning of the placenta and the possibility of the penetration of oxygen and nutrients from the maternal body [46]. Violation of such an exchange between the mother and the fetus may be the reason for a decrease in the body weight of the latter. Such a decrease in weight may be associated with the limited supply of nutrients and oxygen.

The animal neurological testing makes it possible to judge the development and functional state of different levels of the central nervous system. The methods of neurological testing used in this work largely reflect the state of the developing parietal cortex in rat pups. For example, a mature response in the body flip test requires coordination between the vestibular system, muscles, and spine, as well as the input of tactile sensations through vibrissae into the somatosensory cortex [47]. The whisker placing test quantifies the functional state of the sensorimotor cortex and is widely used to assess the severity of damage to this brain region after an experimental stroke or other brain pathologies [48]. Postural correction on an inclined platform also requires the participation of the motor cortex [49] and is a special case of controlling the main posture when the animal is kept on a rotating grid.

In our previous works, it was found that in PHHC rat pups, which had a lag in the formation of neurological reactions, there was a delay in the maturation of nerve cells and increased neurodegeneration in the cerebral cortex and hippocampus at different stages of postnatal ontogenesis compared with the control [3,4]. The large pyramidal neurons in layers V-VI of the parietal cortex were shown to be vulnerable to PHHC as their number was decreased on P20. As is known, the cortical neurons of the fifth layer are the source of the pyramidal tract, which is associated mainly with the activity of the muscles of the distal limbs, especially the hand and fingers [50,51]. Pyramidotomy entails a weakening of the muscle tone of the limbs [52], and muscle effort depends on the number of activated motor units, the synchronism of their action, and the magnitude of the tension of the antagonist muscles. It is likely that the death of large pyramids of layer V of the cortex and a decrease in their number in PHHC pups entails a decrease in the number of activated motor units, which leads to lower performance in tests of body flip and retention on a rotating grid (where the forelimbs play the dominant role), and also in the test of whisker-evoked forelimb placing.

However, it should be noted that all neurological responses tested in this work (body righting, whisker placing, rotating grid) require the simultaneous development of strength and coordination, and a delay in the appearance of these reflexes may also indicate a delay in muscle development and growth.

The delay in the maturation of hippocampal cells, identified in this study, may be the cause of reduced locomotor activity (P16-P18) in PHHC rat pups since, normally, it is at this age that rat pups show the earliest examples of stable adult-like firing found in directional (P16) and place (P16-17) cells [53] in the entorhinal cortex [54] and hippocampus [55], respectively.

Previous clinical studies have suggested that decreased prenatal maternal folate and an increased Hcy level during pregnancy inversely correlated with the performance in the language, visuo–spatial domains, and IQ of six-year-old children [56]. Furthermore, Hcy injection may cause behavioral and morphological changes in prepuberal (P35) and adult (P60) rats, which could be similar to changes found in children diagnosed with attention deficit hyperactivity disorder [57]. In this study, an increase in the dendritic length in dentate gyrus and nucleus accumbens, decreasing the number of mushroom spines in different areas (including nucleus accumbens, CA1), was found in such rats. Our behavioral results are consistent with the data obtained in the morphological study and indicated an increase in neurodegeneration processes and a decrease in the rate of differentiation and maturation of cells, as well as of neurogenesis and synaptogenesis in the parietal cortex and hippocampus during the first month of postnatal life of PHHC rats.

In our study of the neural tissue ultrastructure in the brain cortex and the hippocampus, primary attention was paid to the indicators of the brain tissue maturity: the maturity of the cells, the formation of the neuropil, and the rate of synaptogenesis. The rate of maturation of the cells and neuropile in cortical tissue seems higher than in the hippocampus; the number of axodendritic cortical synapses increases with age both on longitudinally cut dendrites and in glomeruli, which indicates a high rate of formation of neuronal connections. In the hippocampus of rats of the same age we found more intercellular space, numerous growth cones, fewer spines on dendritic branches, and fewer mature synapses than in cortical tissue. Our data suggest PHHC induced the delay in the development of both neocortex and hippocampus.

The literature data indicate that in rat pups affected by various adverse effects during embryogenesis, neuroblast migration can be disturbed in a similar way in different areas of the cortex (prefrontal, temporal, parietal, entorhinal, etc.) since the mechanisms of radial neuroblast migration are generally similar [58]. It is known that a continuous hypoxic condition during brain development affects both the radial [59] and tangential [60] migration of cortical neuroblasts, generating critical alternations in the structure of the cortical plate of newborn rats. The hypoxia on E17 was reported to disturb also the migration of neuroblasts into the hippocampus [61]. We found earlier [34] that prenatal hypoxia at E14 or E18 disrupted cell generation and radial migration and caused crucial alternations in the final arrangement of generated neurons, similar to how it was shown in various non-hypoxic models of prenatal pathology, e.g., in a model of fetal brain damage by ultrasound [62]. The data obtained in the present study of PHHC pups suggest that PHHC resulted in the inability of some neuroblasts to migrate to their correct position in layers V-VI or II-III of the parietal cortex at both E14 and E18, respectively. At the same time, the rate of neuroblast formation in the developing neocortex was reduced. The patterns of the disturbances in neuroblasts migration were shown to be similar in PHHC and the model of prenatal hypoxia. The migration of the cells in the developing brain is known to be dependent on the balance of the various neurotrophins and other signal molecules and transcriptional factors.

Several studies have reported changes in the levels of neurotrophins (NGF, BDNF) in the physiological fluids of the mother and fetus due to the development of pathological pregnancy. Thus, the BDNF level in umbilical cord blood was shown to be lower in premature infants compared to full-term infants; in animals and humans, BDNF and NGF concentrations decrease with infections in the offspring brain and amniotic fluid [21,63,64]. In our study, high levels of Hcy in the mother’s blood [33] also led to a change in the content of the neurotrophic factor BDNF in the fetal brain. Interestingly, there was a significant elevation in the content of only proBDNF on E20, which may be partially explained by an alteration in the *Bdnf* gene expression at this time through a change in the methylation level, as it was shown in another models of HHC (folate deficiency [65] and methionine administration [66]). Unlike E20, on E14 in the group with PHHC, an increase in proBDNF was accompanied by a decrease in mBDNF. Similar changes in the content of the pro- and mature forms of BDNF were found in the study of the acute stress effect on the developing brain of sea bass [67]. An imbalance of proBDNF and mBDNF in the offspring frontal cortex and striatum was also noted in the nicotine mouse model. Interestingly, at the same time, their overall BDNF level had not changed. However, the authors attribute the processing violation to a detectable decrease in the furin content in these structures [68]. 

It is believed that neurotrophin proteolysis is catalyzed by plasmin, furin, and MMPs since the pro-forms of neurotrophins contain sites for their cleavage [69,70]. The suppression of the activity or decrease in the proteases content is characteristic of various neurological and neurodegenerative disorders [69,70,71,72,73]. Thus, in the prenatal stress model, changes in LTP and LTD induction in the hippocampus of animals correlated with an increase in proBDNF levels and a reduction in mBDNF levels with a significant decrease in plasminogen activity and expression [74]. Similar violations of the proteases functioning can occur in rat fetuses with PHHC. Thus, it is believed that proMMP-2, resistant to autoproteolysis, can form under conditions of high levels of Hcy, as well as the inhibition of MMP-2 activation in vitro [75], which contributes to a change in the processing of neurotrophic factors towards the formation of pro-forms. The contribution of this mechanism to the cumulative negative effect of Hcy exposure in vivo requires additional research. However, it corresponds with data obtained in this study, i.e., a decrease in MMP-2 activity on E14. Thus, the change in the processing of BDNF in the fetal brain with PHHC is most likely due to a violation of its gene expression and activity of proteases.

It was noted that vitamin B_12_ deficiency affects the progression of the cell cycle and the differentiation of neuroblastoma cells through the interaction of signaling pathways associated with an increased expression of the *Pp2a*, *Ngf*, and *Tace* genes [76]. In the rat chronic stress model, an increase in *Ngf* mRNA and an inhibition of NGF maturation was detected, followed by an increase in mNGF degradation in the hypothalamus due to MMP [70]. However, in our study, there were no changes in either the protein level or the *Ngf* gene expression in the brain of rat fetuses, which indicates other mechanisms of the high Hcy concentrations effect on developing nerve cells. At the same time, it is especially important to study the effects of the prevailing form since a number of sources mention that the binding of proBDNF to the p75NTR receptor and its mature form to the TrkB receptor have the opposite effect on neuronal cells. While the binding of BDNF to TrkB enhances axon branching, the activation of p75NTR with proBDNF contributes to a decrease in the rate of this process [77]. The action of proBDNF in vivo leads to a decrease in the density of hippocampal spines, as well as the inhibition of synaptic transmissions and a number of other consequences affecting the survival and functioning of neurons [78]. The proBDNF gradient had a suppressing effect on the purified granule cell movement and growth in vitro [15]. Furthermore, in vivo studies have shown that proBDNF acts as a repulsive factor for neuronal cells [15]. Daily exposure to cocaine from E8 until birth can decrease tangential and radial migration of neurons and is associated with a reduction in BDNF levels in the basal forebrain of mice [79]. In this regard, it is very likely that impaired neuroblast migration under conditions of elevated Hcy observed in our study may be partly due to an increase in the fetal brain proBDNF level at both considered time points of prenatal development.

Recent studies indicate that factors that are involved in the development and functioning of the vascular system may also be potential regulators of axonal growth. For example, the vasoconstrictor peptide endothelin-3 acts through the endothelin-A receptor to stimulate axonal elongation [80]. Moreover, several studies have described the expression of three known regulators of the vascular system, namely VEGF receptors (VEGFR1-3), in neuronal cells of the developing nervous system [81,82,83,84]. In vitro studies have reported that VEGF interacts with VEGFR2 (KDR) to promote neurogenesis, neuronal survival, and axonal growth [85]. Conditional deletion of *Vegf* in neuronal progenitor cells resulted in diminished neuronal proliferation, increased apoptosis, and extensive degeneration of the cerebral cortex [86,87]. In contrast, the conditional deletion of *Vegf* from endothelial cells resulted in an abnormal increase in neuronal proliferation and disrupted neuronal migration mouse embryonic cortical development [88]. Despite the fact, that in our study, no changes were found in the expression level and content of VEGF in the whole rat fetal brain, the data obtained on a reduction in the expression of the *Kdr* gene on E14 suggest the involvement of this signaling pathway in the developmental disorder of the fetal brain during PHHC. KDR was shown to be associated with the PlexinD1/Neuropilin-1 receptor complex for Sema3E and become tyrosine-phosphorylated upon SEMA3E activation that is required for the increment in axonal growth [81]. The results of this study also indicate that KDR is involved in axonal wiring through a mechanism dependent on Sema3E and independent of VEGF ligands [81]. Thus, the simultaneous decrease in the expression of *Kdr* and the content of SEMA3E may be one of the mechanisms of impaired brain development in rats with PHHC.

HHC can induce fetal hypoxia [89], and the overexpression of *Bdnf* in neuronal cultures is able to enhance the preconditioning effect of short episodes of hypoxia since there is an expression of antiapoptotic and a decrease in the expression of proapoptotic and proinflammatory genes. At the same time, the knockout of the *Bdnf* gene cancels the above effect and promotes the death of GABAergic neurons, which emphasizes the contribution of the neurotrophic factor to cell survival [90]. It is known that fetal hypoxia can induce an increase in the Caspase-3 activity in the cerebral cortex of newborn animals, as well as Caspase-8 and -9 [91]. In our previous studies, when studying the postnatal effects of PHHC, an elevation of the activity and content of the active form of Caspase-3 was noted, which was accompanied by the neuronal death in the cerebral cortex of rats [3]. The oxidative stress stimulated by HHC can also mediate the development of apoptotic processes [92,93,94]. In the studies of other authors, it is noted that the nervous system formed in early ontogenesis has an increased sensitivity to free radical oxidation and has low antioxidant protection activity [95]. In our previous studies, it was found that maternal HHC caused a decrease in total antioxidant and superoxide dismutase activity and an increase in oxidative modification of DNA together with lipid peroxidation in the brain of newborn rat pups [96]. At the same time, PHHC did not lead to changes in the oxidative modification of proteins in the fetal rat brain on E20, but the level of lipid peroxidation was elevated suggesting oxidative stress activation. Furthermore, in our study, there was a decrease in the level of procaspase-8 in the E20 fetal brain in the group with maternal HHC that, together with Caspase-3 activation, may indicate the development of apoptosis along the internal mitochondrial pathway.

## 5. Conclusions

It can be concluded that PHHC disturbs the mechanisms of early brain development leading to the delay in brain tissue maturation in the neocortex and hippocampus and the formation of the motor reaction of pups in the first month of postnatal ontogenesis.

## Data Availability

The data presented in this study are available on request from the corresponding author.

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
