# Peer review of "Maternal Hyperhomocysteinemia Disturbs the Mechanisms of Embryonic Brain Development and Its Maturation in Early Postnatal Ontogenesis"

_cells, 2023, doi:10.3390/cells12010189_

Round 1

Reviewer 1 Report

This is an interesting study investigating the impact of maternal hyperhomocsyteinmia on offspring health. There are a few revision I think need to be made for the article to be ready for publication.

Firstly, only male offspring were used. I think that this should be reflected in the title of the paper, abstract, and discussion. I think that the authors need to include a statement of why female offspring were not included in study.

I would also ask the authors to include more statistics to be reported in the results section, such as t-statistic and df. I would also like to request that all bar graphs are converted to scatter plot graphs.

For all data how was it analyzed? Was the experimenter blinded to all groups?

Author Response

Reviewer 1

This is an interesting study investigating the impact of maternal hyperhomocsyteinmia on offspring health. There are a few revision I think need to be made for the article to be ready for publication.

A: Many thanks to reviewers for positive and constructive comments on our manuscript. We have addressed reviewer’s comments as listed below and hope that our paper will now satisfy all the requirements for publication in the Cells.

Q1 Firstly, only male offspring were used. I think that this should be reflected in the title of the paper, abstract, and discussion. I think that the authors need to include a statement of why female offspring were not included in study.

A1:The behavior of adult females is unstable due to the instability of the hormonal background associated with the presence of the estrous cycle in females, so males are usually investigated in behavioral experiments, unless the goal is to study females specifically. In our study of the embryonic material we used fetuses of both sexes, but in the study of pups after birth only males was used to exclude the additional factors, that might disturb the PHHC action. The necessary details were added in the manuscript.

Q2 I would also ask the authors to include more statistics to be reported in the results section, such as t-statistic and df. I would also like to request that all bar graphs are converted to scatter plot graphs.

A2: The panels in the figures 3, 7-10 has been now presented as a dot plot charts. The figure 1 wasn’t changed because of the high number of data points on the limited space. Since in fig. 1 which demonstrate the effect of PHHS on body weight and neurobehavioral characteristics in postnatal ontogeny in rats, each of the parts of this figure represents data for two rat groups (control and PHHC) for 8-20 days, it is not possible to convert our graphs into scatterplots.

Q3 For all data how was it analyzed? Was the experimenter blinded to all groups?

A3: All analyses were randomly performed in a blinded manner for all animal groups. According to our practice, in all experiments each animal and sample were given a random number so the experimenters were not aware of the tested groups which were only revealed during the statistical analysis. The procedure has now been explained in the text.

Reviewer 2 Report

Comments to the Authors:

This manuscript tittled: Maternal hyperhomocysteinemia disturbs the mechanisms of embryonic brain development and its maturation in early postnatal ontogenesis” has the propose to show neurodegeneration process induced by hyperhomocysteinemia in embrionary stage. However, there are some important points should be considered and corrected, please attend the following comments.

Mayor Comments:

·       An hyperhomocysteinemia could lead to develop some alterations related to attention deficit disorder. Some reports suggest this could induce neural morphological alterations in prepuber and postpuber stage. I suggest mention this in the discussion to aim correlate the morphological effects induced by an hyperhomocysteinemia. I recommend review: https://doi.org/10.1016/j.jchemneu.2021.102057.

    ·      In figure 2, if you do not measure the EdU in CA1, what is the idea in this figure to present an area of the hippocampus. Did you performed any immunohistochemistry in hippocampus to quantified neuroblasts? Because in the discussion you mentioned that one of the objectives of the study is to demonstrate alterations in neurogenesis, could be important evidence some alterations in neuroblast not only in CA1 even in Dentate Gyrus.

Minor Comments:

·       In Material and methods section, please include the age of the rats used, and if the rats were multiparous or nulliparous.

 ·        I suggest improve the Scheme I, please specify what does it mean: H2O, in the first part of the diagram? because it´s not clear if you refer if the rats received water ad libitum? Even more, please indicate in the scheme, each behavioral tests in the correspondent age.

  ·        In line 230, the authors mentioned they performed a Nissl stain. But, there is not any results derivate of that technique, so, what was the objective to stain with cresyl violet? did you quantified the number of cells? you do not present any figures about it.

Author Response

Reviewer 2

This manuscript tittled: Maternal hyperhomocysteinemia disturbs the mechanisms of embryonic brain development and its maturation in early postnatal ontogenesis” has the propose to show neurodegeneration process induced by hyperhomocysteinemia in embrionary stage. However, there are some important points should be considered and corrected, please attend the following comments.

 A: Many thanks to reviewers for positive and constructive comments on our manuscript. We have addressed reviewer’s comments as listed below and hope that our paper will now satisfy all the requirements for publication in the Cells.

Mayor Comments:

  • Q1An hyperhomocysteinemia could lead to develop some alterations related to attention deficit disorder. Some reports suggest this could induce neural morphological alterations in prepuber and postpuber stage. I suggest mention this in the discussion to aim correlate the morphological effects induced by an hyperhomocysteinemia. I recommend review: https://doi.org/10.1016/j.jchemneu.2021.102057.

A1: Thank you very much for recommending this very interesting work. We mentioned this article in the "Discussion" section

Q2·      In figure 2, if you do not measure the EdU in CA1, what is the idea in this figure to present an area of the hippocampus. Did you performed any immunohistochemistry in hippocampus to quantified neuroblasts? Because in the discussion you mentioned that one of the objectives of the study is to demonstrate alterations in neurogenesis, could be important evidence some alterations in neuroblast not only in CA1 even in Dentate Gyrus.

A2: The areas of the interest in brain cortex and hippocampus, shown on figure 2a presented the areas of the brain studied in electron microscopy as well as the analysis of EdU-labeled cells by fluorescent microscopy. That’s why both parietal cortical area and CA1 of hippocampus were presented in figure 2. The period of the neurogenesis in rat hippocampus began later than in VZ and SVZ of the forebrain and it is prolonged into the postnatal ontogenesis. In the case of the labeling of cells on E14 there were no EdU-labeled cells at all, the number of cells labeled on E18 was rather low and the distribution of data did not permit to obtain reliable conclusion about possibility of the differences between control and PHHC groups. Thus the distribution of the EdU-labeled cells in hippocampus was not included in this study.

Minor Comments:

Q3·       In Material and methods section, please include the age of the rats used, and if the rats were multiparous or nulliparous. 

A3: These details have been provided in the Methods section line 122.

Q4·        I suggest improve the Scheme I, please specify what does it mean: H2O, in the first part of the diagram? because it´s not clear if you refer if the rats received water ad libitum? Even more, please indicate in the scheme, each behavioral tests in the correspondent age.

A4: The scheme was edited in accordance with the Reviewer advice. The first part of the diagram shows a scheme for modeling hyperhomocysteinemia by administering methionine (PHHC group) and the control group of animals, which were additionally injected with water through a probe at the same period of time. This was written in more detail in the section "Materials and Methods" line 140-141. An additional graphical timeline has been included in the Scheme 1b to provide the time points for behavioral studies.

Q5·        In line 230, the authors mentioned they performed a Nissl stain. But, there is not any results derivate of that technique, so, what was the objective to stain with cresyl violet? did you quantified the number of cells? you do not present any figures about it.

A5: According to our common practice the sample of brain slices are subjected to Nissl stain to control the structural anomalies as well as the quality of the studied histological material. As no direct data from Nissl stained material were provided in present work the statement about this method was removed from the text.